# Half a Century of Progress: The Evolution of Wheat Germ-Based In Vitro Translation into a Versatile Protein Production Method

**DOI:** 10.3390/ijms26083577

**Published:** 2025-04-10

**Authors:** Brigitta M. Kállai, Tatsuya Sawasaki, Yaeta Endo, Tamás Mészáros

**Affiliations:** 1Department of Molecular Biology, Semmelweis University, Tűzoltó u. 37-47, H-1094 Budapest, Hungary; kallai.brigitta@semmelweis.hu; 2Proteo-Science Center, Ehime University, 3 Bunkyo-cho, Matsuyama 790-8577, Japan; sawasaki@ehime-u.ac.jp; 3Ehime Prefectural University of Health Sciences, 543 Takooda, Tobe-cho 791-2101, Iyo-gun, Japan; endoyaeta@gmail.com

**Keywords:** cell-free protein synthesis, in vitro translation, wheat germ extract, high-throughput protein synthesis, eukaryotic protein expression, extracellular proteins, membrane proteins

## Abstract

The first demonstration of wheat germ extract (WGE)-based in vitro translation synthesising a protein from exogenously introduced messenger ribonucleic acid (mRNA) was published approximately fifty years ago. Since then, there have been numerous crucial improvements to the WGE-based in vitro translation, resulting in a significant increase in yield and the development of high-throughput protein-producing platforms. These developments have transformed the original setup into a versatile eukaryotic protein production method with broad applications. The present review explores the theoretical background of the implemented modifications and brings a panel of examples for WGE applications in high-throughput protein studies and synthesis of challenging-to-produce proteins such as protein complexes, extracellular proteins, and membrane proteins. It also highlights the unique advantages of in vitro translation as an open system for synthesising radioactively labelled proteins, as illustrated by numerous publications using WGE to meet the protein demands of these studies. This review aims to orientate readers in finding the most appropriate WGE arrangement for their specific needs and demonstrate that a deeper understanding of the system modifications will help them make further adjustments to the reaction conditions for synthesising difficult-to-express proteins.

## 1. Early Days

Cell-free protein synthesis (CFPS) has become an indispensable research tool of molecular biology since its first description using rat liver cell extract in 1955 [1]. Mammalian cell-free translation was soon followed by the establishment of *E. coli* protein extract-based CFPS, indicating that cell independence and robustness are general features of the translation apparatus in all organisms [2]. The subsequent development of in vitro translation, i.e., the production of proteins encoded by exogenous messenger ribonucleic acids (mRNAs), proved instrumental in deciphering the genetic code and understanding the molecular mechanisms of protein biosynthesis [3,4].

In addition to its importance in elucidating previously hidden aspects of the translation process, the potential to synthesise proteins of interest by introducing exogenous mRNAs into cell-free extracts suggested that in vitro translation is also a reasonable approach to protein production. This possibility was investigated, with a number of organisms and cell lines being considered as potential sources of translation machinery for protein-producing systems. These included *E. coli*, wheat germ, rabbit reticulocytes, yeast, insect cells and mammalian cell lines such as HeLa and Chinese hamster ovary (CHO) cells, with the first three being the most widely used eukaryotic translation systems over the years [5,6,7,8]. All cell-free translation systems have their own benefits and shortcomings; therefore, careful consideration should be given when selecting the appropriate system for specific research applications [9,10]. Some of the advantages of the wheat germ extract (WGE) in vitro translation system are related to the ease of obtaining the necessary starting material, the relatively low levels of endogenous mRNA and nuclease activity and the high protein synthesis capacity of dormant embryos [11]. These latter characteristics obviate the requirement for ribonucleic acid elimination, making the production of cell extracts cost-effective and straightforward. A further beneficial characteristic of the wheat germ translation machinery is its relatively low sensitivity to the codon bias, making the codon optimisation of heterologous protein-coding genes unnecessary [12]. The objective of this review is to provide a comprehensive and up-to-date overview of the advancements in WGE in vitro translation, which have led to the development of an efficient and versatile protein synthesis approach (summarised in Table 1).

## 2. Increasing Productivity

Although the original wheat germ CFPS demonstrated a prolonged activity compared to the *E. coli*-based in vitro translation, its productivity did not meet the requirements of an effective protein-producing platform. It was postulated that the low productivity of WGE was attributable to certain proteins of the endosperm, which contaminate the traditionally purified WGE. These proteins encompass the ribosome-inactivating RNA N-glycosidase tritin and other translation inhibitors, such as thionin, in addition to a battery of nucleases and proteases [44,45,46]. This hypothesis was corroborated by the observation that careful selection of wheat embryos and their subsequent extensive washing significantly enhanced the translation activity. The use of WGE from selected and thoroughly washed embryos resulted in an in vitro translation system that was active for four hours, as opposed to the one-and-a-half-hour reaction time observed in the original system. In addition, sucrose density gradient analysis of translation reactions derived from washed embryos showed the formation of large polysomes, indicating high protein synthesis activity [11].

Eukaryotic heterogenous nuclear RNA (hnRNA) undergoes a complex process before being exported from the nucleus as mature mRNA. The modifications of nascent mRNA result in the formation of the apt protein-coding open reading frame, while also providing essential elements of standard eukaryotic translation initiation, such as 5′ methylguanosine capping and poly(A) tail formation [47]. To initiate translation, the mRNA must be bound to the small ribosomal subunit by a protein complex that includes the cap-binding eukaryotic initiation factor 4E (eIF4E) protein and the tail-binding poly(A) binding protein. Achieving the two required RNA modifications in vitro is technically challenging. While the cap structure can be made using capping enzymes that incorporate the modified free dinucleotide at the 5′ end, the efficiency of the reaction is variable, and without further purification, an excess of free modified nucleotide remains in the reaction mixture, which also binds to eIF4E, thus inhibiting translation [14]. The construction of 100–250-nucleotide-long poly(A) tail-encoding vectors is only a theoretical option, as the repetitive nucleotides increase the instability of the vectors. Therefore, the construction of translation vectors with this motif is usually not a reasonable approach to provide a tail to the in vitro-transcribed mRNA; alternatively, a segmented poly(A) tail can be used [48]. The poly(A) tail can also be added post-transcriptionally to the in vitro-transcribed mRNA, but this approach requires a further reaction after transcription.

To circumvent the obstacles associated with the generation of translationally active mRNA, the tobacco mosaic virus (TMV) Ω translation enhancer sequence was employed for replacement of the 5′ cap. The Ω element is known to recruit the HSP101 heat shock protein, which in turn recruits the eukaryotic initiation factor 4G (eIF4G) initiation factor to facilitate translation [49]. It was also shown that extending the 5′ end of the Ω sequence with the GAA nucleotide triplet resulted in a further increase in translation efficiency, reaching approximately 75% of that observed for capped mRNA. An additional benefit of using this modified mRNA is that its effective concentration range is wider than that of the capped mRNA, thus avoiding the need for meticulous optimisation of the mRNA concentration in in vitro translation reactions [14].

The potential to design a poly(A) tail replacing the 3′-untranslated region (UTR) with a short poly(A) repeat was hinted at by the finding that the tail is dispensable for translation in yeast [50]. The yeast data were corroborated by the findings that translation in WGE is not contingent on any specific sequence but is solely dependent on the length of the 3′-UTR. These results led to the construction of the bacteriophage SP6 promoter-driven, WGE-optimised pEU3b vector [14]. Contrary to this observation, the 3′-UTR sequence has been shown to influence translation efficiency. It has been demonstrated that 3′-UTRs with multiple stem-loops form a more compact structure than a random coil polymer, bringing the mRNA ends closer together and enhancing translation efficiency [51]. In addition, shorter 3′-UTR sequences have been identified, which promote the closed-loop formation of mRNA without a 5′ cap and poly(A) tail [52,53,54]. The pEU3-NII variants of the pEU3b vector are based on the bacteriophage T7 transcription system and have been constructed to streamline the creation of vectors encoding proteins of interest and the purification of synthesised proteins [55]. A panel of these vectors enables seamless cloning of coding regions of proteins of interest by using a ligation-independent cloning site that was inserted into the original vector. Purification of the synthesised proteins is aided by a variety of tobacco etch virus (TEV) protease cleavable affinity tags, including 6xHis, tandem 6xHis, glutathione S-transferase (GST), Halo, AviTag, FLAG and N-terminal GST combined with C-terminal 6xHis/AviTag [56,57].

The previously described pEU3b vector has been modified by replacing the Ω sequence with artificial translation enhancers. As an open system, in vitro translation is an optimal choice for in vitro evolution-based oligonucleotide selection; thus, it was used to isolate the best translation enhancer element by ribosome display from a pool 73 nucleotides long of random sequences, which excluded guanine to avoid premature AUG start codons in the 5′-UTR of the mRNA. Two of the identified oligonucleotides were found to possess a similar translation enhancer capacity to those of the Ω sequence and were used to create the pEU-E01/E02 vectors by replacing the TMV translation enhancer element of the original pEU3b plasmid [15]. The development of another artificial enhancer was driven by the need for an efficient wheat germ-based coupled in vitro transcription/translation system, and its design was inspired by the cricket paralysis virus (CrPV) internal ribosome entry site (IRES) [58,59]. SP6 RNA polymerase efficiently synthesises mRNAs with the 5′-GAA methylguanosine cap substitution, but its activity declines sharply at lower temperatures, which are optimal for the wheat germ translation machinery. Although T7 RNA polymerase remains active at lower temperatures, it requires transcripts starting with the GGG trinucleotide to achieve full activity. The conflicting requirements of transcription and translation were met by using the mutant variant of the above IRES, which was found to be 3–4 times more productive than the canonical translation of mRNA transcribed using the pEU-E01 vector [33].

Protein synthesis is the most energy-demanding process of the cell [60]. In addition to the production of amino acids and ribosomes, the coupling of amino acids to their pertinent transfer ribonucleic acid (tRNA) and the diverse steps of translation also require molecules of macroergic bonds, such as adenosine 5′-triphosphate (ATP) and guanosine 5′-triphosphate (GTP). Consequently, a constant supply of energy is a critical element of any effective in vitro translation reaction [61]. Supplementing the reaction with higher concentrations of ATP is not cost effective and results in an excess of adenosine 5’-diphosphate (ADP) by-product, which competes with ATP for enzyme binding sites. Although multistep pathways have also been developed to maintain the ATP pool of in vitro translation, the most commonly used approaches to energy regeneration rely on substrate-level phosphorylation of ATP [62]. The high-energy phosphate bond required for this reaction is provided by either phosphoenolpyruvate or acetyl phosphate in *E. coli* extract and creatine phosphate in wheat germ-based in vitro translation [13,63,64]. Recycling of guanosine 5’-diphosphate (GDP) by transferring the γ-phosphate of ATP can be achieved by ribosome-associated nucleoside diphosphate kinases (NDKs) [65,66]. Recently, simultaneous ATP/GTP regeneration was achieved by a *Cytophaga hutchinsonii* polyphosphate kinase that uses polyphosphates as substrates [67].

Polyamines are known to be stably associated with ribosomes, stimulating translation elongation and reducing translation misreading. In accordance with this, early studies in cell-free protein synthesis reported the stimulation of translation and a reduction in the Mg^2+^ requirement when spermidine or spermine was added to the reaction mixture. Consequently, polyamines have become standard components of most in vitro translation systems [68].

The first generation of batch format cell-free translation systems was limited in its use for preparative protein production due to the systems’ short lifetime of less than one hour. This limited reaction time can be attributed to the exhaustion of substrates and energy-providing components of the translation, on the one hand, and the production of translation-inhibiting by-products, on the other hand. A variety of arrangements have been devised with the objective of enhancing the efficacy of protein production by maintaining a continuous supply of amino acids and energy components to the translation mixture, collectively designated as the feeding buffer. The initial solution, known as continuous-flow cell-free (CFCF), employed a micro-ultrafiltration tube with an 8 kDa cut-off, through which the feeding buffer was pumped in order to be delivered to the translation mixture holding tube and to eliminate the small-molecular-weight by-products [13]. In addition to its longer lifetime and orders of magnitude higher productivity, another advantage of the CFCF is that it can facilitate the downstream purification step, as there is no significant leakage of the protein-synthesising apparatus, while the translated protein of interest can be collected from the flow-through fraction if a higher cut-off membrane is used [69,70]. Due to the clear advantages of CFCF, such as high productivity and the convenient separation of synthesised proteins, diverse variations in it have been published [18]. However, it has not become a popular in vitro translation system of research laboratories because they require relatively complex instrumentation and it is clogged easily with precipitation-prone proteins. To circumvent the limitations of the CFCF arrangement while maintaining its high productivity, continuous-exchange cell-free (CECF) systems have been developed. In this configuration, the translation mixture and the feeding buffer are separated by a semipermeable membrane and the two solutions are exchanged passively through diffusion. The first reported CECF employed *E. coli* extract to conduct a coupled transcription/translation reaction and produced 1.2 mg/mL protein in 14 h [71]. The prokaryotic cell extract-based CECF was soon followed by the WGE-based system, demonstrating the general applicability of this configuration. In terms of protein synthesis capacity, the WGE-based CECF outperformed the first reported CFCF, being active for up to 60 h and yielding 4 mg of target protein in a 1 mL reaction when supplemented daily with fresh mRNA [11]. While the CECF was found to be an effective approach to in vitro translation and relatively straightforward to set up, it did not fully meet the requirements of a high-throughput protein production system. Driven by the motivation to create a cell-free protein synthesis system that is amenable to high-throughput protein studies, a solution was sought to address the issue of continuous supply of translation mixture without the formation of a physical barrier between the reaction and translation solutions. This goal was achieved by developing a bilayer in vitro translation method that exploits the high density of the translation mixture. In this configuration, the translation mixture is simply placed underneath the feeding buffer, a task that can be conveniently performed by pipetting robots. The lack of membrane separation of the two solutions does not appear to significantly reduce productivity; the bilayer system demonstrated sustained activity for over 10 times longer than batch-mode reactions and yielded sub-mg/mL amounts of protein [16]. Although the CECF remains the most widely used approach for the synthesis of milligram quantities of proteins, the bilayer configuration has become the method of choice for small-scale protein production for high-throughput protein studies [31]. Recently, an optimised batch-mode WGE-based CFPS was presented with comparable productivity to the bilayer arrangement, termed coupled in vitro transcription/translation (cIVTT) and applied for the assembly of gene regulatory cascades using protein-responsive artificial riboswitches [34,35]. Alternative approaches, such as protein-producing hydrogels (P-gel), in which the protein-coding linearised plasmids are covalently crosslinked to a gel matrix, could be also considered as a means of producing proteins with higher efficiency and yield, but this system demands further development for routine applications [25].

## 3. High-Throughput Protein Analysis

One of the earliest applications of the bilayer format provides an excellent illustration of the capabilities of this system [24]. The authors of the paper set out to achieve a highly ambitious goal, namely the conversion of the human transcriptome into the proteome. To this end, they first produced a Gateway entry vector library that covered approximately 70% of human genes by using various sources of complementary DNA (cDNA) and used them to create a destination vector library of different tags. The resulting library was used in polymerase chain reactions to provide DNA templates for in vitro transcription to produce the open reading frame (ORF) encoding mRNAs. Prior to commencing high-throughput protein synthesis, WGE was compared to *E. coli* CFPS, and *E. coli* and mammalian cell-based protein expression systems, and the data collected showed that the WGE bilayer format outperformed all other approaches analysed. In view of these results, high-throughput protein synthesis was implemented using the WGE-based bilayer system. Next, 13,364 entry clone-derived mRNAs were added to the translation reaction, resulting in the successful synthesis of 12,996 proteins. The translation mixtures were separated into soluble and insoluble fractions by centrifugation, and the results indicated that 12,682 of the expressed proteins were soluble. Furthermore, according to secondary structure prediction, 3040 proteins of the soluble fraction contained at least one single transmembrane domain. To evaluate whether the solubility of the proteins was accompanied by their correct folding, the enzyme activity of 75 expressed phosphatases was measured and 77% were found to be active, indicating that WGE cell-free translation produces the majority of proteins in their correct conformation. The authors also determined the productivity of the bilayer format and showed that a 150 µL volume reaction produced an average of 2.6 µg of soluble protein, which is sufficient for most functional protein studies. Finally, the authors constructed a protein microarray by expressing 13,277 N-terminally GST-tagged human proteins. The total translation mixtures were directly printed on a glass slide using DNA microarray apparatus, and the expressed proteins were detected by fluorescently labelled GST-specific antibody. Although only 238 spots were not decorated with the antibody, the fluorescence signal of the remaining spots exhibited a considerable range, indicating that the yield of the different proteins is highly variable. According to a recent publication, this shortcoming can be alleviated by saturating the GST-coated spots of the glass plates with FLAG-GST-labelled proteins [72]. The resulting protein-modified glass plates, called A-cubes, held 240 spots and were used to identify autoantibodies in patients with systemic sclerosis and polymyositis/dermatomyositis.

The microarray results demonstrate that the direct application of in vitro translation mixtures in high-throughput studies without the purification of the protein of interest is severely limited by the substantial variability in the yields of expressed proteins. To ensure the use of equal amounts of proteins, it is necessary to purify and quantify the expressed proteins prior to the downstream experimental steps. Although it has been demonstrated that GST-tagging and magnetic glutathione beads can be used to purify thousands of expressed proteins in parallel, high-throughput protein purification remains a challenging task [73]. A number of publications have addressed this challenge by eliminating the need for protein purification and combining WGE with the use of AlphaScreen and AlphaLISA (ALPHA for Amplified Luminescent Proximity Homogeneous Assay) for identification of protein–protein and protein–DNA interactions [42,74,75,76,77,78]. The ALPHA assay is a bead-based, no-wash, homogenous assay that relies on a donor bead, which converts ambient oxygen to the singlet state upon excitation with monochromatic light and an acceptor bead that emits a fluorescence signal when it comes into contact with the singlet oxygen. This approach has been commercialised, thus both types of beads are available with a variety of coatings for binding antibodies and the most commonly used affinity tags. In addition to hypothesis-driven studies, the combination of WGE and ALPHA is also a rational choice for holistic protein interaction analysis, as demonstrated by the generation of protein arrays for autoantibody screening [75,77]. Target proteins were produced and biotinylated simultaneously by adding BirA, a sequence-specific biotin ligase and biotin to both the translation mixture and feeding buffer of the bilayer format in vitro translation in 384-well plates. After overnight incubation, 1 μL of translation solution was added to 0.05 μL of synovial tissue extract or 0.025 μL of serum, followed by the addition of AlphaScreen streptavidin donor beads and Protein G acceptor beads, and the interactions were determined by measuring the intensity of the luminescence signal [75,77].

The combination of WGE in vitro translation and AlphaScreen was proven to be a rational method for drug screening as well, as evidenced by the identification of novel abscisic acid (ABA) and gibberellin (GA) analogues. All 14 ABA receptors of *Arabidopsis thaliana* were produced with biotin labelling and combined with the FLAG-tagged abscisic acid insensitive 1 (ABI1), a known interacting partner of ABA receptors, prior to AlphaScreen measurement. The luminescence signal collected confirmed the previously described interactions, demonstrating that some of the receptors form complexes with ABI1 in an ABA-independent manner, while others require the plant hormone for the interaction. Subsequently, a chemical library comprising 9600 compounds was screened based on the interactions between pyrabactin resistance 1 (PYR1), one of the ABA receptors, and ABI1. This resulted in the identification of a novel ABA receptor agonist compound, designated JFA1 (julolidine and fluorine-containing ABA receptor activator 1). The new receptor agonist was observed to suppress seed germination, increase the expression of ABA-inducible genes and enhance drought tolerance in *Arabidopsis thaliana* plants. These findings suggest that this approach allows for the large-scale screening of agonist or antagonist compounds for plant hormones [79]. The phytohormone GA is widely used in agricultural contexts to promote a range of processes, including germination, stem elongation, flowering, fruit maturation and the production of seedless fruits; therefore, there is a constant demand for new GA analogues [80]. The authors of a recent publication sought to identify novel GA receptor agonists for an agriculturally important crop, the grape, rather than conducting their studies on model plants of phytology such as *Arabidopsis thaliana*. Their approach leveraged the GA-dependent interaction of the GA receptor, GID1a (GA-INSENSITIVE DWARF1), and the nuclear growth repressor proteins, called DELLA (aspartic acid–glutamic acid–leucine–leucine–alanine) proteins, and followed the rationale described above in screening a library of 9,600 small molecules. Having identified the most potent interaction-promoting molecule, diphegaractin, they also demonstrated its activity in lettuce, rice and *Arabidopsis*. Finally, diphegaractin was shown to have GA-like biological activities in agricultural crops such as lettuce and mandarin, suggesting that the data provided by the applied in vitro molecular targeted drug screening can be very effectively translated into practical application-oriented studies [81]. Thalidomide was a popular sedative for pregnant women in the middle of the last century but was soon withdrawn from the market because it caused severe embryopathies. Thalidomide and its derivatives bind to the cereblon protein (CRBN), a key molecule in thalidomide-induced teratogenesis, resulting in an altered binding specificity of CRBN [82,83]. Yamanaka et al. [84] succeeded in identifying previously unknown thalidomide-dependent substrates of CRBN using WGE-translated CRBN and the Human Transcription Factor Protein Array (HuTFPA) consisting of over a thousand proteins. AlphaScreen-based screening of human transcription factor substrates of CRBN with thalidomide on HuTFPA resulted in the identification of six transcription factors that bind to CRBN in a thalidomide-dependent manner, three of which were also confirmed by biochemical assays. In addition, thalidomide treatment of chicken embryos reduced the protein level of the identified promyelocytic leukaemia zinc finger protein (PLZF), proving the credibility of the in vitro data obtained [84].

Recently, a workflow has been developed for the automated construction of vectors and the cell-free production of plant proteins using *E. coli* and wheat germ lysates for synthetic biology approaches. The system contains acceptor vectors with a panel of N- and C-terminal tags for the detection, purification and improved expression of proteins, as well as an acoustic liquid handling platform. The authors demonstrated that functional assays can be performed without the need for protein purification, thereby significantly increasing the throughput of experiments [41].

## 4. Protein Complexes

The vast majority of intracellular proteins are part of dynamically changing protein complexes that enable cells to increase the efficiency of metabolic modules, fine-tune signalling pathways and respond adequately to both extracellular and intracellular stimuli. Therefore, proteins often need to form complexes to be active or to achieve their proper functionality, and studying them in isolation may not reveal their true roles or may even lead to wrong conclusions. Certain complexes may be reconstituted from recombinant proteins produced separately, but formation of many multiprotein complexes demand coexpression of a number of proteins, and cotranslational folding of complexes on ribosomes can assist in the proper folding of the interacting partners [85,86].

Similarly, the protein complexes generated by the cell-free system can be obtained through the separate translation or cotranslation of the proteins of interest. The separate production method was employed to identify the cyclin-dependent kinase subunits of *Arabidopsis thaliana* cyclin B1. The authors synthesised the proteins by WGE and, following the mixing of the translation reactions, pulled down the protein complexes by immobilised metal affinity chromatography (IMAC) via the hexahistidine labelling of cyclin. The interactions were demonstrated by Western blot analysis and an in vitro kinase assay of the purified complexes [87]. The cotranslation of interacting partners can be beneficial in the context of pull-down experiments. Our observations have revealed that the solubility of *Arabidopsis thaliana* gamma-tubulin, a protein prone to polymerisation in vitro, is significantly improved when it is coexpressed with its GST-tagged interacting partner *Arabidopsis thaliana* E2 promoter binding factor A (AtE2FA), as opposed to GST (personal communication).

The DNA of all eukaryotes is packaged into nucleosomes by wrapping two copies of each of the histones H2A, H2B, H3 and H4 around approximately 147 bp of DNA [88]. Beyond this basic structure, the many variants of the canonical histones assemble different nucleosomes, conferring unique properties on the nucleosomes that impact chromatin stability and, in turn, affect DNA replication and repair and transcriptional regulation [89]. Far from being an inert packaging structure, chromatin is a dynamic scaffold capable of responding appropriately to specific stimuli to regulate DNA accessibility. This rapid adaptation is mainly achieved by post-translational modifications of the histones of the nucleosomes [90]. In vitro reconstruction of chromatin using defined protein components is a valuable tool to describe the potential histones of different nucleosomes and identify the possible post-translational modifications of histones. The basic requirement for in vitro chromatin assembly is the preparation of histones in the correct stoichiometry, which is a laborious task using either the nucleus or recombinant protein overexpressing bacterial cells as a source of histones. In recent years, WGE has proven to be a sensible solution to this challenge [36,37,38,91].

## 5. Extracellular Proteins

In vivo, protein synthesis—apart from the translation of a dozen proteins encoded by mitochondria—takes place in the reductive environment of the cytosol, which provides optimal conditions for the translation machinery and folding of most proteins. However, about 9% of the proteins in the human proteome are secreted into the oxidative extracellular space, and a considerable number of membrane proteins possess extracellular domains. These proteins cannot fold into their native spatial structure in the cytosol [92]. The challenge of correctly folding these proteins is met by targeting them to the lumen of the endoplasmic reticulum, which provides an isolated chemical environment and a battery of proteins that aid in the proper folding of proteins and the formation of correct disulfide bridges [93].

Traditional WGEs have been optimised for increased translation efficiency and do not contain organelles, so the buffer of the system is reductive and lacks enzymes responsible for disulfide bridge formation. These conditions make the production of disulfide bridge-holding proteins in this system a delicate issue. Despite these shortcomings, WGE has been shown to be suitable to produce such proteins. The first published attempt to synthesise proteins with appropriately formed disulfide bridges showed that a single-chain antibody variable fragment (scFv) could be produced in functional form in dithiothreitol (DTT)-free buffer and in the presence of exogenously added protein disulfide isomerase (PDI), as demonstrated by the antigen-binding capacity of in vitro-translated scFv. It is noteworthy that the removal of DTT resulted in a reduction in protein synthesis efficiency, which was further exacerbated by the addition of PDI [17]. To establish a more efficient disulfide-forming WGE, the addition of quiescin sulfhydryl oxidase (QSOX)—an endoplasmic reticulum resident, FAD-binding, disulfide-introducing protein—to the translation mix was tested in the production of the murine found in inflammatory zone 1 protein (FIZZ1) [94]. FIZZ1 is a small secretory protein with 10 cysteines that must form five matching disulfide bonds to become functional and able to suppress Th2 cytokine expression in splenocytes [95]. The data obtained showed that the reaction supplemented with QSOX yielded more soluble FIZZ1 than those supplemented with either PDI or a combination of QSOX and PDI. The authors suggested that these results may indicate that the added QSOX1 and the endogenous PDI of wheat germ work together in the correct folding of FIZZ1. Activity measurements of the synthesised proteins highlighted the importance of correct disulfide bridge formation; although the soluble form of FIZZ1 produced in the absence of QSOX had no free thiols, only FIZZ1 synthesised in the presence of QSOX was able to reduce interleukin production by splenocytes [96]. The same research group followed a similar rationale to synthesise a biologically active small scorpion α-toxin from *Androctonus australis hector* (AahII) with four non-consecutive disulfide bridges, after failing to produce it in a soluble format even with the disulfide bridge optimised SHuffle™ T7 Express *E. coli* cells. Expression of AahII either with the GST or hexahistidine tag or without the tag in WGE resulted in production of only the GST-tagged AahII, while the other two variants could not be translated. Contrary to the previous publication, it was found that neither the addition of QSOX nor PDI increased the productivity of the translation reaction [27]. Importantly, neither FIZZ1 nor AahII were produced using DTT-depleted reaction mixtures and feeding solutions during in vitro translation. The coronavirus disease 2019 (COVID-19) pandemic sparked interest in the synthesis of proteins from the disease-causing virus, and one research group used WGE to produce the spike protein of severe acute respiratory syndrome coronavirus 2 (SARS-CoV-2). They used a commercially available, DTT-depleted, modified version of the WGE that included endoplasmic reticulum oxidoreductase-1 α (ERO1α) and PDI (called EP-WG system) to ensure proper disulfide bridge formation of the spike protein [42]. PDI is regenerated by ERO1α by oxidising the reduced PDI [97]. Comparison of the productivity of the conventional and modified WGE showed that the latter system could produce more than twice as much spike protein as the regular in vitro translation approach. Functional measurements of the spike protein produced by the two different systems also demonstrated the superiority of the EP-WG system, as the protein synthesised by this approach had an order of magnitude lower dissociation constant with the angiotensin-converting enzyme 2 (ACE2) receptor than the spike protein produced by the conventional WGE [42].

In view of the results of the above publications, it is reasonable to assume that the redox conditions of the WGE should be optimised if disulfide-carrying proteins are to be produced on a larger scale. It has been described that the production of human and mouse prion-like Doppel protein, mouse interleukin-22 and virus-like particles can be enhanced by fine-tuning the redox condition of the *E. coli* lysate translation mixture by finding the most appropriate ratio of reduced to oxidised glutathione (GSH:GSSG), but similar experiments have not yet been performed with WGE [98,99]. In addition to establishing the appropriate redox conditions, it has also been shown that blocking the free sulfhydryl groups of proteins from *E. coli* extracts by iodoacetamide pre-treatment promotes the correct folding of proteins containing disulfide bridges, an approach not investigated in WGE but worth considering for application [100].

The tobacco (*Nicotiana tabacum*) Bright Yellow-2 (BY-2) cell lysate-based in vitro translation system is capable of synthesising proteins with disulfide bonds without the addition of extra enzymes and tuning the redox conditions of the standard system. This phenomenon is most likely due to the presence of native, actively translocating microsomal vesicles derived from the endoplasmic reticulum and Golgi [101]. Therefore, the addition of microsomes to the WGE is expected to produce proteins with appropriately formed disulfide bridges if a targeting signal is fused to the N-terminus of the protein, and addition of exogenous signal recognition particle (SRP) can further improve translocation efficiency [102]. Traditionally, the thoroughly characterised, commercially available canine pancreatic microsomes have been the method of choice to supplement the cell-free translation mixture, but it has been shown that cultured insect cells, which were isolated by a simplified protocol that does not remove SRP from the membranes, can also be used as a source of microsomes [103]. For ethical reasons and because of the relatively consistent quality and easy availability of cell cultures, cell culture-derived microsomes meet the requirements of an ideal in vitro translation system and should therefore be the preferred approach for supplementing WGE with microsomes.

## 6. Membrane Proteins

The expression of membrane proteins (MPs) in living systems is a challenging process, mainly due to their hydrophobic nature and the need for a specific chemical milieu, membrane platform and chaperone system to facilitate their correct folding. In eukaryotes, these requirements are met by the endoplasmic reticulum network of the cells, whereas in prokaryotes, MPs are inserted directly into the cell membrane in conjunction with complexes such as SecYEG. Overloading the MP-synthesising capacity of cells generally leads to the formation of inclusion bodies and disruption of cell membranes, both of which are detrimental to cells. These challenges underscore the potential value of CFPS as a promising approach for MP production. As open systems, the CFPS reactions can be supplemented with components that enhance solubility and facilitate the folding of these proteins. Gentle lysis of some mammalian, insect and plant cell lines can ensure the presence of artificial derivatives of the endoplasmic reticulum, called microsomes, in the extract prepared for in vitro translation, making these systems directly applicable to MP synthesis [59,103]. However, many protein extracts from other sources, including WGE, contain very few microsomes per se and therefore need to be modified for MP production. Below we discuss the modifications made to the conventional WGE CFPS system that resulted in the successful synthesis of MPs.

Some of the MPs with only one or two transmembrane domains (TMDs) can be efficiently recovered from the soluble fraction without supplementing the standard WGE translation mixture with additives, as demonstrated by the production of the *Plasmodium falciparum* type 1 integral membrane protein, PfAMA1, for immunisation purposes in malaria vaccine research [104]. The correct folding of these proteins is thought to be aided by the endogenous phospholipid content of WGE, estimated to be 725 mM [105].

In most cases, however, the production of MPs is a much more arduous task and requires careful optimisation of in vitro translation. Depending on the additives used, the MPs synthesised by CFPS can be broadly divided into two categories: those based on the addition of detergents followed by lipid reconstitution (D-CF mode), and those based on the addition of membrane-mimicking lipid structures—such as liposomes, polymersomes, nanodiscs or microsomes—directly to the translation mixture (L-CF mode) [106]. It should be noted that this classification is more theoretical than practical, given that the translation mixture synthesising MPs often contains both types of supplementations.

The incorporation of spontaneously micelle-forming detergents represents the technically less challenging approach to the production of MPs. The micelles are in the close proximity to the ribosomes, thus the translated proteins need not to be transported to membranes; they directly form proteomicelles, which prevents the precipitation of non-water-soluble proteins. One of the earliest demonstrations of MP production with bilayer format WGE using detergents was performed by Nozawa et al. [21]. The authors tested the expression of *Arabidopsis thaliana* phosphoenolpyruvate/phosphate translocator 1 (AtPPT1) with the addition of a panel of detergents and found that the various polyoxyethylene alkyl-ether derivatives (Brij) and digitonin were the most effective in increasing the soluble fraction of the target protein, with Brij 35 being selected for further experiments [21]. A recent review summarising the detergents used in the WGE system showed that non-ionic detergents are generally more suitable for increasing the solubility of MPs [107]. However, in many cases, they also have been observed to inhibit translation. Therefore, it is important to remember that while several detergents confer beneficial effects in terms of increasing solubility, they can also significantly reduce translation efficiency and the protein yield achieved [107]. Furthermore, in certain cases, the type and concentration of detergent used should be optimised separately for the translation mixture and the feed buffer, as demonstrated for the chromophoric bacteriorhodopsin [105]. Taken together, these data suggest that screening is highly recommended to identify the appropriate detergent prior to the large-scale synthesis of the protein of interest.

In addition to utilising detergents, the authors of the aforementioned publication by Nozawa et al. [21] also investigated the reconstitution of MPs into liposomes simultaneously with the translation. In the initial experiment, AtPPT1 produced in the presence of Brij 35 was combined with liposomes prepared from asolectin following the in vitro translation reaction. In the subsequent experiment, the translation of AtPPT1 was performed using a reaction mixture and feeding buffer that was supplemented with both Brij 35 and liposomes. Measurement of the transport activity of proteoliposomes derived from the two arrangements demonstrated the superiority of the latter, which exhibited a 140-fold higher activity than the proteoliposomes reconstituted after in vitro translation [21]. Interestingly, this large difference in activity between the two approaches was not reflected in the level of association of AtPPT1 with proteoliposomes, i.e., approximately half of the synthesised protein was found in the proteoliposome fraction with both post- and cotranslational membrane integration. This suggests that cotranslational integration into a membrane structure promotes the correct folding of MPs and increases the yield of functional MPs produced by CFPS [26]. Encouraged by these results, the same research laboratory set out to assess the general applicability of liposomal supplementation of WGE for MP synthesis in the bilayer format without the addition of detergents. The study investigated the cotranslational association of an array of mammalian MPs with liposomes and demonstrated that lipid/MP complexes were formed for all 40 proteins tested, which belonged to different families and had a wide range of transmembrane domains (from 1 to 14). Cotranslational MP association with liposomes was quantified after isolation by density gradient ultracentrifugation and ranged from 43 to 73%. The lowest degree of association with liposomes was observed for proteins with a single transmembrane domain, most probably due to the proportionally small hydrophobic region, which may limit the effectiveness of the interaction [26]. Nomura et al. demonstrated that the cytochrome b5 (cytb5) could be synthesised in the bilayer format directly into ~3–4 mm liposomes, called giant liposomes, made from a mixture of cholines, and that cytb5 could be used as a “hydrophobic tag” to display a hydrophilic fusion protein on the liposome surface [108]. In another experiment, the formation of a two-protein composed active complex between cytb5 and human stearoyl-CoA desaturase 1 (hSCD1) was also achieved by the addition of soy lecithin-derived liposomes and ascorbate-stabilised Fe^2+^ to the reaction chamber of the dialysis translation system [109]. Although the desaturase activity of hSCD1 was slightly increased upon addition of hemin, the WGE extract contained sufficient hemin to produce active cytb5. Furthermore, the authors demonstrated that post-translational recombination of the proteoliposomes with individual proteins, or cotranslation of mRNAs encoding both proteins in the same reaction mixture, is equally efficient in terms of the activity of the complex formed. Notably, endogenous wheat germ proteins, including Hsp70 (70 kilodalton heat shock protein), EF1α (elongation factor 1 alpha) and a 16.9 kDa heat shock protein, were also found to be associated with the translated MPs, implying a potential chaperone role for these proteins in the process [109]. In addition to the bilayer and dialysis formats, the successful incorporation of MPs into liposomes in the absence of detergents was also shown in the batch format. The reaction mixture was supplemented with large unilamellar liposomes prepared from egg yolk l-α-phosphatidylcholine, and the resulting proteoliposomes were used in a rapid filtration assay to demonstrate the dehydroascorbate transport activity of human glucose transporter member 10 (GLUT10) [110].

The choice of the membrane-mimicking lipid type for optimal protein translation is often determined by the specific characteristics of the expressed protein. The mitochondrial inner membrane ADP/ATP carrier (AAC) protein containing two TM segments was translated into a functionally active form by WGE through spontaneous integration into liposomes containing a blend of biomimetic lipids, with the cardiolipin content enhancing the translation rate and integration efficiency [111]. Another illustrative example is the expression of the wild-type human ether-a-go-go-related gene (hERG) potassium channel for use in a drug screening platform. hERG was first translated into liposomes consisting of phosphatidylcholine, phosphatidylethanolamine and cholesterol, and subsequently incorporated into a mechanically stable artificial bilayer lipid membrane (BLM) using a solvent-free system and a silicon chip as a membrane support, while maintaining hERG enzymatic activity [112,113].

CFPS can also be adopted for the translation of more structurally challenging MPs, such as G protein-coupled receptors (GPCRs) [114]. GPCRs possess intra- and extracellular loops connecting seven transmembrane helices, with a highly conserved disulfide bridge between the second extracellular loop and the third transmembrane domain, which is crucial for the GPCR structure and function. While functional expression of some GPCRs can sometimes be achieved by conventional WGE translation and co- or post-translational incorporation into artificial membranes, in most cases, the utilisation of a further modified WGE system is necessary. Combining the strengths of the previously discussed bilayer and dialysis methods of translation has led to the development of the “bilayer–dialysis method”, which has resulted in significant improvements in MP yields, including GPCRs. Comprehensive protocols for this technique have been published in the literature [29,30]. In this methodology, the liposome-containing bilayer reaction is set up in a dialysis cup, which is then immersed in the translation buffer. This arrangement allows the translation reaction to be fed and by-products to be removed both from above at the phase boundary and from below through the dialysis membrane. This approach led to the successful expression of 25 GPCRs in sufficient quantities for immunisation studies [28]. A more thorough investigation revealed a shortcoming of GPCR expression with liposome-completed WGE translation: the model protein studied, human dopamine D1 receptor (DRD1) showed the topology of the receptor was bidirectional due to the inability to control the directionality in the absence of the Sec translocon. As a result, the functional receptor with ligand-binding capability represented only 0.02% of the total amount of receptor synthesised [28]. A similar divergent membrane insertion pattern was observed by Ritz et al. during expression of olfactory receptor 1 (Olfr1) into dioleoylphosphatidylcholine (DOPC) and soybean phosphatidylcholine (PC) liposomes, resulting in undetectable ligand binding of the produced receptor [115]. In contrast, the integration of the cOR52 olfactory receptor into asolectin liposomes was confirmed by a ligand-specific response of the receptor, but the unidirectionality of membrane incorporation was not investigated by other means in this case [116]. Despite the aforementioned limitations, the results obtained substantiate the rationale for optimising of the WGE-CFPS for the expression of olfactory receptors to be utilised in the promising field of electronic olfactory systems (bioelectronic noses) [117].

Subsequent improvements in yield and functional GPCR content were achieved by using ~200 nm glycerosomes prepared from asolectin and 20% glycerol instead of regular liposomes. Although the presence of glycerosomes resulted in a slight reduction in protein translation efficiency, the correct integration of the human histamine H1 receptor (HRH1) into the membrane was increased to 0.1% [118,119]. A further advancement combined the use of asolectin glycerosomes with the Systemic Evolution of Ligands by Exponential Enrichment (SELEX) method for aptamer selection and was termed “proteoliposome-SELEX” [120]. The technique was successfully employed to select aptamers, which inhibited Mas-related G protein-coupled receptor X2 (MRGPRX2)-associated histamine release in mast cells [121].

Although the introduction of glycerosomes slightly increased the proportion of correctly oriented MPs, the levels achieved remain far from optimal. Another plant extract, *Nicotiana tabacum* Bright Yellow-2 (BY-2) cell lysate, contains native microsomes derived from the endoplasmic reticulum and Golgi apparatus, allowing the integration of MPs without the need for exogenous lipids or detergents. Notably, the lysate successfully produced the GPCR cannabinoid receptor type 2 (CB2) in a functional form as measured by G-protein activation rates. Interestingly, according to the authors, CB2 was targeted to microsomal membranes by a passive mechanism as well, bypassing the microsomal translocation pathway typically regulated by the melittin signal peptide (MSP) [101]. These unexpected results are consistent with earlier studies using canine pancreas-supplemented WGE for in vitro translation [122]. Although our knowledge of the mechanism underlying passive microsomal targeting in this system remains limited, it merits further investigation.

In addition to liposomes composed of natural phospholipids, polymersomes formed from amphiphilic block copolymers have emerged as artificial and more stable vesicle alternatives that mimic cell membranes [123]. Nallani et al. [124] incorporated the integral membrane protein claudin-2, which engages in cell–cell interactions, into poly(butadiene)-poly(ethylene oxide) (PBD-PEO) polymersomes. The binding of claudin-2-specific antibodies to the proteopolymersomes was confirmed using surface plasmon resonance measurements [124]. The same research group followed a similar approach for the functional incorporation of the dopamine receptor D2 (DRD2) and light-harvesting complex II (LHCII). LHCII was incorporated into spherical polymersomes loaded with 0.4 M trehalose, thereby enhancing their stability for transmission electron microscopy (TEM) studies, and into planar polymer bilayers modified with thiolated lipoic acid for surface plasmon resonance analysis [125,126]. Another effective strategy to achieve stable immobilisation of polymer-stabilised MPs for label-free binding analysis is to couple streptavidin on chips and mix a small fraction of biotinylated lipids into the polymersome membranes [127].

Nanodiscs composed of membrane scaffold proteins and lipids have also attracted attention as a more stable MP-producing system compared to liposomes, as they can also provide a native-like environment for MPs, thereby supporting their stability and function. For example, Li et al. demonstrated that the *Arabidopsis thaliana* regulator of G protein signalling 1 (AtRGS1) exhibited comparable activity when translated in nanodiscs as opposed to unilamellar liposomes [128]. The study revealed that a sequential expression strategy, in which nanodiscs were self-assembled by first adding the mRNA encoding the membrane scaffold protein 1D1 (MSP1D1) and 0.6 mM cardiolipin, followed by the addition of the mRNA encoding the target protein AtRGS1, provided more AtRGS1 than simultaneous translation of the two proteins. This phenomenon can be explained by the fact that the pre-synthesised nanodiscs aided proper folding, thereby increasing the solubility of AtRGS1 [128]. Recently, efforts have been made to optimise a system for the in vitro production of artificial rubber particles capable of replicating the synthesis of natural rubber using nanodisc-supplemented WGE, but the approach still needs to be developed to make it suitable for in vitro rubber production, as it is currently only adapted to the synthesis of small polyisoprene chains [129,130].

## 7. Protein Synthesis for Structural Studies

A fundamental requirement for the structural analysis of proteins is the availability of an ample amount of properly folded protein of interest. The most straightforward and cost-effective method of producing proteins for structural studies is provided by the bacterial cell expression systems, due to their simplicity and cost-effectiveness. However, many eukaryotic proteins, including MPs, glycoproteins, disordered proteins and toxic proteins, are often not readily or not expressible in prokaryotic systems or in living cells. Historically, CFPS has not been considered suitable for structural biology analyses; however, recent enhancements to the WGE-based CFPS have positioned it as an effective technique to produce proteins that are challenging to express in conventional systems, in sufficient amounts for structural investigations (examples of such proteins are listed in Table 2).

Recent advancements in solid-state nuclear magnetic resonance (NMR) analysis, including increased magic-angle spinning frequencies and dynamic nuclear polarisation, along with the impact of cryogenic electron microscopy (cryo-EM) on protein structural analysis, have led to a consistent decrease in the amount of protein required for three-dimensional protein structure determination [131,132,133,134]. However, compared to other protein studies, the protein yield requirements for structural analysis remain relatively high and represent a significant part of the cost of these studies. Therefore, it is advisable that users initiate with small-scale expression tests before scaling up production. As outlined by Fogeron et al. [107], supplementation of translation mixtures with additives such as ions, detergents and lipids has been shown to increase protein yield and solubility. In addition to determining the optimal buffer and temperature conditions, application of an appropriate protease inhibitor has also been demonstrated to elevate the productivity of the in vitro translation. Following the determination of the optimal concentration of potassium acetate and magnesium acetate, Noirot et al. [135] showed that reducing protein degradation by adding the protease inhibitor E-64 resulted in higher yields of the yeast prion protein Ure2p. The importance of identifying the apt protease inhibitor was highlighted by the finding that phenylmethylsulfonyl fluoride (PMSF) completely inhibits translation [135]. In terms of productivity, it is important to note that the use of algal amino acid mixtures for labelling should be avoided in WGE-based expression, as these mixtures have been observed to decrease the expression yield [136]. The strategies outlined in the previous sections for optimising expression conditions should also be considered.

Although both coupled and uncoupled transcription–translation formats are generally employed in WGE-CFPS, the latter is usually preferred for structural biology applications because it allows for the fine-tuning of the protein expression conditions without interfering with the mRNA transcription step. Nevertheless, it has been demonstrated that high incorporation efficiencies of selenomethionine and [*U*-^15^N] labels for X-ray crystallography and NMR spectroscopy, respectively, can be achieved even in the coupled format [137,138].

Compared to cell-based protein expression systems, one of the main advantages of the optimised WGE is its low endogenous mRNA content, which allows isotope-labelled amino acids (^2^H, ^13^C, ^15^N) to be incorporated almost exclusively into the protein encoded by the exogenous mRNA, resulting in a homogeneously labelled sample. Furthermore, the scrambling phenomenon, where labelled amino acids are enzymatically converted into unwanted metabolites, is also low in cell-free extracts and can be further reduced by supplementing the reaction with scrambling inhibitors [139]. The beneficial effect of using scrambling inhibitors has been demonstrated in the production of proteins by stereo-array isotope labelling (SAIL) in WGE. Unwanted exchange reactions at backbone and side-chain positions were reduced by including the aspartate transaminase inhibitor aminooxyacetic acid and the glutamine synthetase inhibitor L-methionine sulfoximine in the translation mixture to produce *Chlorella* ubiquitin [140]. Further supplementation with an additional alanine transaminase inhibitor, β-chloro-L-alanine, along with soybean phosphatidylcholine, allowed the characterisation of the TF_o_*c* subunit oligomers of bacterial F_o_F_1_-ATP synthase by solid-state NMR [141,142]. A comprehensive review of protein labelling strategies for solution NMR can be found in Hoffmann et al. [139].

The analysis of protein structures necessitates the utilisation of pure proteins; consequently, the selection of an affinity tag, which facilitates the cost-effective and efficient isolation of the protein of interest, is a critical consideration. The workflow for protein synthesis by WGE and purification for electron microscopy and protein crystallography, as presented by Novikova et al., employed the 3xFLAG affinity tag [39]. While it is possible to achieve a high level of purity in a single chromatography step using this tag, the approach is limited by the high cost of the 3xFLAG peptide required for elution. An improvement to this system was provided by Sugihara et al. [143] with the publication of His-tagged 3xFLAG peptide expression in *Brevibacillus choshinensis*. The produced peptide can be used to elute 3xFLAG-tagged proteins and subsequently removed from the solution by IMAC, thus minimising excess peptide, which could interfere with downstream analysis of the protein [143]. An alternative peptide-based affinity tag, Strep-tag II, provides a similar or even higher level of purity of protein from WGE but at a significantly lower cost since desthiobiotin is used for elution. Numerous studies have demonstrated the efficacy of this tag, thus suggesting it as the optimal choice for the isolation of proteins from WGE for structural studies [107,144,145,146,147,148,149,150], although in some cases, the presence of Strep-tag II, similar to other tags, has been shown to affect the structure under investigation [151]. Whilst providing lower-level purity, the most generally applied affinity tag, the His-tag, can also be used; however, the DTT of regular translation buffers must be removed prior to using IMAC if non-DTT-compatible beads are applied. Another option is to use DTT-free translation buffers, but this may reduce the efficiency of the translation reaction. To achieve a high purity of certain tagged proteins, commercially available, special WGEs can be used, which have been pre-cleared on the respective affinity resin prior to the translation reactions. However, we found that the activity of a zinc finger-containing ubiquitin ligase was lower when translated with IMAC pre-cleared WGE, presumably due to the removal of an endogenous wheat germ protein required for its activity (personal communication). In addition to the isolation of proteins by affinity purification, it is also advisable to remove the putative protein-interacting nucleic acids, which may interfere with structural analysis, by means of benzonase treatment [148].

Some research laboratories have established WGE-based platforms for high-throughput structural analysis of proteins. In the protein-producing pipeline at the Center for Eukaryotic Structural Genomics (CESG), small-scale translation reactions are initially set up in dialysis cups and incubated overnight to assess the success of expression and purification. Large-scale reactions involve the translation of the proteins of interest in a dialysis bag at 26 °C for 48 h in the presence of [*U*-^13^C,^15^N]-labelled amino acids. The replacement of the substrate solution at 12 h intervals resulted in a labelling efficiency of >95% [19,20]. Using this approach, a multitude of protein structures have been determined by solution NMR (Table 2).

Scientists from the Molecular Microbiology and Structural Biochemistry at the University of Lyon have developed a 4–6-week workflow for solution and solid-state NMR studies of viral assemblies, using a protein production protocol established by Takai and colleagues [31,32]. Their translation process utilises a dialysis cassette with a 10 kDa molecular weight cut-off in dialysis mode to achieve a higher protein concentration than in the bilayer arrangement. The incubation is conducted at 22 °C for 16–20 h, with 60–70 rpm shaking and inclusion of a mild detergent in case of MPs. For instance, it has been demonstrated that the duck hepatitis B virus (HBV) small (S) envelope protein can spontaneously self-assemble following expression in WGE to form subviral particles, while the HBV capsid protein Cp183 can form capsid assemblies and the assembly can be modulated with the addition of capsid assembly modulators (CAMs) [136,152,153]. The same group, using the bilayer configuration, identified the major and minor phosphorylation sites of the intrinsically disordered PreS domains of the HBV large (L) envelope protein by interpretation of NMR chemical shifts. Notably, both the full-length HBV L protein and its PreS fragments co-purified with endogenous WGE Hsc70, a heat shock protein that is likely involved in the cytosolic anchorage of PreS, underscoring the efficacy of WGE in viral mechanism studies [145,154]. The laboratory also showed that the NMR spectra of the cytosolic domain of the Crimean–Congo haemorrhagic fever virus (CCHFV) glycoprotein n (Gn) could be measured after increasing the stability of the translated protein through supplementation with ZnSO_4_, thereby ensuring the correct zinc finger formation [147].

The first crystal structure of a WGE-produced enzyme was determined by X-ray diffraction in 2007 (Table 2). This enzyme was the *Pab*I restriction endonuclease from *Pyrococcus abyssi*, the synthesis of which would otherwise be toxic to a living organism. Another advantage of the WGE was that it facilitated the efficient incorporation of selenomethionine, enabling crystal structure determination by single-wavelength anomalous diffraction. Uncoupled transcription and translation was used to prevent digestion of the DNA plasmid used as a template for mRNA transcription, and translation was carried out at in the bilayer format at 26 °C for 16 h [22]. In 2010, a detailed protocol for the bilayer production and crystallisation of proteins for X-ray studies was published [23].

At the Environmental Molecular Sciences Laboratory, a multiscale protein production pipeline was set up for electron microscopy and X-ray structural analysis. This workflow includes four different scales of reaction (MINI, MIDI, MAXI and MEGA) depending on the experimental purpose and is carried out in the bilayer format for at least 20 h at 15 °C [39]. The open nature of the system allowed the expression of a heterocomplex composed of a pseudoenzyme, pyridoxal 5′-phosphate synthase-like subunit PDX1.2, involved in the regulation of B_6_ biosynthesis in plants, and its cognate enzyme PDX1.3. The PDX1.2/PDX1.3 protein assemblies of varying stoichiometries were achieved by applying different DNA template molar ratios of both partners in the transcription reactions, followed by their coexpression. This research also validated previous observations according to which cotranslation of interacting partners is often indispensable to achieve functional proteins [40].

A cell-free protein crystallisation (CFPC) method for protein structure determination by X-ray diffraction using miniaturised, small-scale reaction mixtures in the dialysis mode has recently been developed by scientists at the Tokyo Institute of Technology [43]. In a proof-of-concept experiment, nanocrystals of the bacterial crystalline inclusion protein A (CipA) were formed directly in the translation mixture, eliminating the need for purification. The formation of twin crystals was inhibited by the addition of polyethylene glycol 400 (PEG 400), and the formation of larger, high-quality crystals was promoted, presumably due to liquid–liquid phase separation, which is thought to be a crystallisation-promoting factor [43]. In another experiment, Kojima et al. showed that crystals of the intrinsically disordered protein c-Myc and its eight mutants could be obtained in as little as 72 h from a small-scale reaction in a dialysis cup [155]. The method used a polyhedrin monomer as a scaffold crystal, which had previously been shown to crystallise rapidly in the cell-free system [43].

**Table 2 ijms-26-03577-t002:** Examples of 3D structural models of proteins synthesised using WGE-CFPS.

Name	Description	Source Organism	Translation Method	Structure Determination Method	PDB/BMRB (#) Identifier	Ref.
MUB1	membrane-anchored ubiquitin-fold protein 1 (At3g01050)	*Arabidopsis thaliana*	dialysis	solution NMR	1SE9	[19]
MAPR2	membrane-associated progesterone-binding protein 2 (At2g24940.1)	*Arabidopsis thaliana*	dialysis	solution NMR	1T0G	[156]
DRLB1	dynein light chain roadblock-type 1	*Mus musculus*	dialysis	solution NMR	1Y4O	[157]
AIG2LA	putative gamma-glutamylcyclotransferase AIG2-like protein A (At5g39720.1)	*Arabidopsis thaliana*	dialysis	solution NMR	2G0Q	[158]
*Pab*I	restriction endonuclease-type II Pab1 domain-containing protein	*Pyrococcus abyssi*	bilayer	X-ray diffraction	2DVY	[22]
STR16	thiosulfate sulphurtransferase 16 (At5g66040.1)	*Arabidopsis thaliana*	dialysis	solution NMR	1TQ1	[159]
F24M12.60	oxidised thioredoxin AtTrx *h*1 (At3g51020.1)	*Arabidopsis thaliana*	dialysis	solution NMR	1XFL	[160]
ZNF593	C2H2-type zinc finger protein 593	*Homo sapiens*	dialysis	solution NMR	1ZR9	[161]
SNX22	sorting nexin-22	*Homo sapiens*	dialysis	solution NMR	2ETT	[162]
UBL3	ubiquitin-like protein 3	*Homo sapiens*	dialysis	solution NMR	2GOW	[163]
MLP28	major latex protein-like protein 28 (At1g70830.1)	*Arabidopsis thaliana*	dialysis	solution NMR	2I9Y	[164]
OARD1	apo ADP-ribose glycohydrolase OARD1 (apo C6orf130)	*Homo sapiens*	dialysis	solution NMR	2LGR	[165]
DCN1L	DCUN1 domain-containing protein	*Galdieria sulphuraria*	dialysis	X-ray diffraction	3KEV	[166]
NS5A-D1D2D3	nonstructural protein 5A domains	Hepatitis C virus	bilayer	solution NMR	#26702	[167]
NS5A-AHD1	nonstructural protein 5A AH-linker-D1	Hepatitis C virus	bilayer	solid-state NMR	#50380	[150]
PDX12/PDX13	pyridoxal 5′-phosphate synthase-like subunit PDX1.2/PDX1.3 heterocomplex (At3g16050, At5g01410)	*Arabidopsis thaliana*	bilayer	cryo-EM	7LB6	[40]
Artn	molecular chaperone artemin	*Artemia franciscana*	bilayer	cryo-EM	7RVB	[168]
PhC	polyhedrin	*Bombyx mori* cypovirus 1	bilayer	X-ray diffraction	7XHR 7XWS	[43]
CipA	crystalline inclusion protein A	*Photorhabdus luminescens*	dialysis	X-ray diffraction	7XHS	[43]
Gn	glycoprotein n cytosolic domain	Crimean–Congo haemorrhagic fever virus	bilayer	solution NMR	#52372	[147]
S-HDAg	antigen, small isoform	hepatitis delta virus	dialysis	solid-state NMR	#52512	[151]
c-Myc/PhC	c-Myc proto-oncogene protein/polyhedrin	*Homo sapiens/Bombyx mori* cypovirus 1	dialysis	X-ray diffraction	8J2Q 8WLG 8X8S 8X8V	[155]
NMR: nuclear magnetic resonance; cryo-EM: cryogenic electron microscopy						

## 8. Outlook

Since the publication of the first WGE-based CFPS more than fifty years ago, the approach has undergone significant development, resulting in a substantial increase in productivity. Current systems achieve mg/mL yields that are several magnitudes higher than the original. Furthermore, the various optimisations have rendered the WGE-based CFPS system suitable for synthesising difficult-to-express proteins, such as membrane and extracellular proteins, and protein complexes. The broad applicability of these enhanced systems is evidenced by numerous publications spanning fields from high-throughput functional protein analysis to protein structure determination. It is therefore anticipated that it will continue to serve as a valuable tool of protein research in the years ahead.

## Figures and Tables

**Table 1 ijms-26-03577-t001:** Milestones in wheat germ extract (WGE)-based cell-free protein synthesis (CFPS).

Year	Milestone	Improvement	Ref.
1973	Batch in vitro translation using wheat germ extract	Translation of exogenous viral and eukaryotic messenger ribonucleic acid (mRNA) proceeded for up to 1.5 h.	[5,6]
1988	Continuous-flow cell-free (CFCF) translation	The continuous, active flow of feeding buffer through the ultrafiltration-separated reaction mixture extended the reaction time to 40 h, transforming the WGE into a preparative-scale CFPS.	[13]
2000	Improvement of wheat germ extract	Careful selection of wheat embryos and thorough washing increased translation efficiency by removing ribosome-inactivating proteins and other translational inhibitors.	[11]
2002	Improvement of the mRNA used for translation	The introduction of a 5′ cap replacing GAAΩ or GAA-artificial translation enhancer sequences, along with a long 3′-untranslated region (UTR) and a short poly(A) repeat, achieved translation efficiency comparable to capped mRNAs. These innovations streamlined the mRNA generation process for WGE in vitro translation.	[14,15]
2002	Bilayer method for translation	Continuous feeding of translation without a physical barrier enabled small-scale protein production for high-throughput studies and simplified large-scale production in 6-well plates.	[16]
2003	Supplementation with exogenous protein disulfide isomerase (PDI)	Functional synthesis of a single-chain antibody variable fragment with proper disulfide bonds using dithiotreitol-free translation buffer and exogenous PDI.	[17]
2004	Continuous-exchange cell-free (CECF) translation	The passive diffusional exchange of the feeding solution through a semipermeable membrane resulted in a simpler setup compared to the CFCF system, while maintaining comparable efficiency.	[18]
2004	Protein-producing pipeline for structural analysis of proteins by solution nuclear magnetic resonance (NMR)	Small-scale translation reactions in dialysis cups, followed by large-scale reactions in dialysis bags, resulted in labelling efficiency >95%.	[19,20]
2007	Detergent-supplemented WGE (D-CF mode) for membrane protein synthesis	The inclusion of an appropriate detergent increased the soluble fraction of the *Arabidopsis thaliana* multi-pass membrane protein phosphoenolpyruvate/phosphate translocator 1 (AtPPT1).	[21]
2007	Structure determination by X-ray crystallography	The first crystal structure determined using WGE-produced protein demonstrated that the approach enabled the incorporation of selenomethionine into the cell-toxic *Pab*I restriction endonuclease.	[22,23]
2008	High-throughput protein synthesis	Using the transcriptome as a template, 13,277 human proteins were produced on a whole-proteome scale with nearly 100% efficiency. This achievement led to the creation of protein microarrays, dubbed the ‘human protein factory’, for applications in genomics research, drug discovery and other fields.	[24]
2009	Protein-producing hydrogel (P-gel) system for translation	The incorporation of the DNA template into a hydrogel resulted in at least 10 times higher expression yields in a coupled in vitro transcription/translation system than in solution-based systems.	[25]
2011	Liposome-supplemented WGE for membrane protein synthesis (L-CF mode)	A bilayer setup of producing functional lipid/membrane protein complexes resulted in a >40% association rate of membrane proteins with 1 to 14 transmembrane domains in liposomes, making it suitable for large-scale membrane protein production.	[26]
2013	Supplementation with exogenous quiescin sulfhydryl oxidase (QSOX)	The murine found in inflammatory zone 1 (FIZZ1) cysteine-rich secreted protein was synthesised in a physiologically functional spatial structure with proper disulfide bridges.	[27]
2015	Bilayer-dialysis method for membrane protein synthesis	The bilayer translation system was implemented using a dialysis cup containing liposomes, which was immersed in the feeding buffer. This setup efficiently produced several G protein-coupled receptors (GPCRs).	[28,29,30]
2017	Protein-producing pipeline for structural analysis of proteins by solid-state NMR	Optimised high-yield expression of membrane proteins in a detergent-solubilised form, followed by their high-level purification by Strep-tag II-based affinity chromatography and reconstitution in lipids with low lipid/protein ratios.	[31,32]
2021	Coupled in vitro transcription/translation (cIVTT) in batch and dialysis format	The application of a mutant variant of the cricket paralysis virus (CrPV) internal ribosomal entry site (IRES) significantly increased productivity and performed effectively in supergiant unilamellar vesicles (SGUVs).	[33,34,35]
2021	De novo and histone chaperone-mediated assembly of nucleosomes	Reconstitution of nucleosomes from canonical and variant histones using a positioning DNA sequence resulted in chromatin assembly devoid of typical epigenetic modifications, making it suitable for chromatin studies.	[36,37,38]
2021	Protein-producing pipeline for structure determination by cryogenic electron microscopy (cryo-EM)	The open format of the CFPS allowed precise stoichiometric control of protein coexpression and thus the assembly of pyridoxal 5′-phosphate synthase-like (PDX) heterocomplexes for the first cryo-EM structure determination of a WGE-produced complex.	[39,40]
2021	Workflow for automated DNA assembly and cell-free expression	Miniaturised cell-free reactions were performed by using an acoustic liquid handling platform for high-throughput functional assays without the need for protein purification.	[41]
2022	Supplementation with exogenous endoplasmic reticulum oxidoreductase-1 α and protein disulfide isomerase (EP-WG system)	Synthesis of disulfide-bonded receptor-binding domain of severe acute respiratory syndrome coronavirus 2 (SARS-CoV-2) spike protein.	[42]
2022	Cell-free protein crystallisation (CFPC) method	Production of high-quality crystals directly in the translation mixture.	[43]

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
