# Peer review of "Half a Century of Progress: The Evolution of Wheat Germ-Based In Vitro Translation into a Versatile Protein Production Method"

_ijms, 2025, doi:10.3390/ijms26083577_

Round 1
Reviewer 1 Report
Comments and Suggestions for Authors
This review is well-written, provides strong scientific insights, and is well-structured. Its objective is clear and aligns with the scope of the International Journal of Molecular Sciences. Therefore, we highly recommend this review for publication. Here are some minor comments:
The abstract needs to reflect the overall structure of the manuscript.
Many sentences need suitable references, so please check this issue through the manuscript.
Please check the full names of all abbreviations when they first mention them.
Please write italics in vitro throughout the manuscript.
Line 20, please avoid expressions like “ we hope,…”
Please replace “The beginnings” with “ Introduction”.
Line 51, please insert a suitable reference after the word “embryos”, the same in line 85 after the word “translation”.
Line 503, it seems that this paragraph is linked with the former one.
Author Response
Summary:
We sincerely thank Reviewer 1 for the constructive feedback and positive evaluation of our manuscript. Below, we provide detailed responses to each comment and outline the corresponding revisions made to the manuscript. All changes have been highlighted in the revised version using track changes.
Comment 1:
The abstract needs to reflect the overall structure of the manuscript.
Response:
Thank you for this suggestion. We have revised the abstract to better reflect the structure of the manuscript by summarising key sections, including advancements in WGE-based in vitro translation, high-throughput applications, and challenges in synthesising difficult-to-express proteins. The updated abstract can be found on page 1, lines 12–27.
Comment 2:
Many sentences need suitable references, so please check this issue through the manuscript.
Response:
We have carefully reviewed the manuscript and added appropriate references where necessary.
Comment 3:
Please check the full names of all abbreviations when they first mention them.
Response:
We have reviewed the manuscript to ensure that all abbreviations are defined upon first use.
Comment 4:
Please write italics in vitro throughout the manuscript.
Response:
We have ensured that in vitro is italicised consistently throughout the manuscript. Similarly, in vivo and de novo has also been italicised where applicable.
Comment 5:
Line 20: Please avoid expressions like “we hope,…”.
Response:
We have replaced subjective language such as "we hope" with more objective phrasing. One page 1, Line 24, "This review aims to".
Comment 6:
Please replace “The beginnings” with “Introduction”.
Response:
Thank you for the suggestion but we prefer to keep a non-canonical structure. The section title has been updated to "Early days" to better reflect the story.
Comment 7:
Line 51: Please insert a suitable reference after the word “embryos”, the same in line 85 after the word “translation”.
Response:
References have been added as follows:
- After "embryos" on page 2, line 54, we added reference 11.
- After "translation" on page 4, line 91, we added reference 15.
Comment 8:
Line 503: It seems that this paragraph is linked with the former one.
Response:
Thank you for pointing this out, there was indeed an error and have merged this paragraph with the preceding one for better flow. The revised text can be found on page 13, line 513.
Additional Clarifications
While reviewing the manuscript based on reviewer comments, we also added further sentences to better explain certain concepts or findings. These additions are highlighted using track changes in the revised document for transparency.
Thank you again for your valuable feedback!
Reviewer 2 Report
Comments and Suggestions for Authors
Wheat germ extract (WGE)-based in vitro translation has evolved significantly since its first use over fifty years ago, leading to higher productivity and the development of high-throughput protein production systems. These advancements have made WGE-based translation suitable for synthesizing challenging proteins, such as membrane and extracellular proteins, as well as protein complexes. This review explores the modifications made to the system and provides examples of its applications in protein studies, aiming to help researchers optimize conditions for synthesizing difficult-to-express proteins.
Some minor points:
1) DOI numbers should be added for the following references: 25, 35, 40, 82, 91, 92, 98, 114, 120, 135, 146.
2) The authors should check the DOI numbers for references 3, 32, 45, 54; provided DOI numbers do not correspond to the cited papers.
3) The authors' names and paper titles should be written in lowercase in the references (references 1, 3, 6 etc.).
4) The year appears twice in references 7, 49, 51, 53, 59, 61.
5) For reference 32, some of the details are missing.
6) Throughout the paper, the authors should italicize "in vitro" (lines 25, 32, 34 etc.) and "in vivo" (line 358).
7) Recommendation: for better clarity, the authors could present a table with summarized data, clearly displaying the different generations of WGE CFPS systems, their innovations, and improvements in productivity.
Author Response
Summary:
We thank Reviewer 2 for the insightful comments and suggestions. We have carefully addressed each point and revised the manuscript accordingly. All changes are highlighted using track changes in the revised document.
Comment 1:
DOI numbers should be added for the following references: 25, 35, 40, 82, 91, 92, 98, 114, 120, 135, 146.
Response:
We have added DOI numbers for these references.
Comment 2:
The authors should check the DOI numbers for references 3, 32, 45, and 54; provided DOI numbers do not correspond to the cited papers.
Response:
We have verified and corrected these DOIs.
Comment 3:
The authors' names and paper titles should be written in lowercase in the references (references 1, 3, and 6 etc.).
Response:
We have corrected the authors' names.
Comment 4:
The year appears twice in references (7, 49 etc.).
Response:
This duplication has been corrected for all affected references.
Comment 5:
For reference 32, some of the details are missing.
Response:
Missing details for reference 32 (now reference 68) have been added.
Comment 6:
Throughout the paper, italicize in vitro (lines 25 etc.) and in vivo (line 358).
Response:
Both in vitro and in vivo are now italicised consistently throughout.
Comment 7:
Recommendation: Present a table summarizing WGE CFPS systems and their innovations.
Response:
We have included a summarising table (Table 1) that highlights the milestones in WGE CFPS systems along with their key innovations and improvements. This table can be found on pages 2–4, and we hope that it aligns with the Reviewer's suggestion.
Additional Clarifications
While reviewing the manuscript based on reviewer comments, we also added further sentences to better explain certain concepts or findings. These additions are highlighted using track changes in the revised document for transparency.
Thank you again for your valuable feedback!